# Self-Ensemble of $N$-best Generation Hypotheses by Lexically Constrained Decoding

**Ryota Miyano[†], Tomoyuki Kajiwara[‡], Yuki Arase[†]**
[†]Graduate School of Information Science and Technology, Osaka University
[‡]Graduate School of Science and Engineering, Ehime University
[†]{miyano.ryota, arase}@ist.osaka-u.ac.jp
[‡]kajiwara@cs.ehime-u.ac.jp

## Abstract

We propose a method that ensembles $N$-best hypotheses to improve natural language generation. Previous studies have achieved notable improvements in generation quality by explicitly reranking $N$-best candidates. These studies assume that there exists a hypothesis of higher quality. We expand the assumption to be more practical as there exist *partly* higher quality hypotheses in the $N$-best yet they may be imperfect as the entire sentences. By merging these high-quality fragments, we can obtain a higher-quality output than the single-best sentence. Specifically, we first obtain $N$-best hypotheses and conduct token-level quality estimation. We then apply tokens that should or should not be present in the final output as lexical constraints in decoding. Empirical experiments on paraphrase generation, summarisation, and constrained text generation confirm that our method outperforms the strong $N$-best reranking methods.

## 1 Introduction

While the beam search is one of the most common decoding methods in natural language generation, it suffers from the beam search curse (Koehn and Knowles, 2017; Yang et al., 2018; Ott et al., 2018; Stahlberg and Byrne, 2019) where a large beam size degrades the quality of generation. As a remedy to this problem, previous studies explored better alternatives from $N$-best hypotheses (Fernandes et al., 2022) as represented as *reranking* and minimum Bayes decoding (Müller and Sennrich, 2021; Eikema and Aziz, 2022), which only modify the decoding procedures. There are two types of reranking approaches. Discriminative methods train rerankers to predict specific evaluation metric scores of each hypothesis (Shen et al., 2004; Bhattacharyya et al., 2021; Lee et al., 2021). In contrast, generative methods use generic rerankers that have been used for other purposes, such as language models (Yee et al., 2019; Ng et al., 2019).

Different from methods that involve computationally expensive model training such as the minimum risk training (Müller and Sennrich, 2021; Eikema and Aziz, 2022), these ranking-based methods are efficient and easily applicable to trained models.

Nonetheless, these reranking methods assume that there is a single hypothesis of higher quality in the $N$-best, which may not be practical depending on the generation model and also inputs. Therefore, we enhance the assumption; there should be candidates that are *partly high-quality* but may be imperfect as the entire sentences. Our method identifies and merges these higher-quality fragments to derive a high-quality output using lexically constrained decoding (Lu et al., 2022). Specifically, our method trains a token-level quality estimator that predicts whether a token in a hypothesis should be or should not be included in the final output. It then uses the quality estimation (QE) results of the $N$-best hypotheses to compose positive and negative lexical constraints and generates the final output using the generation model.

As a contribution of this study, we propose the $N$-best ensembling method for improving the quality of language generation, which is easy to apply to a variety of language generation tasks. Empirical experiments on paraphrasing (Takayama et al., 2021), summarisation (See et al., 2017; Hermann et al., 2015; Narayan et al., 2018), and constrained text generation (Lin et al., 2020) confirm that our assumption holds and the proposed method outperforms strong reranking-based methods.

## 2 Preliminary: Lexically Constrained Decoding

Lexically constrained decoding has been employed in various language generation tasks to apply task-specific knowledge on generation, e.g., for machine translation using a bilingual dictionary of technical terms as constraints (Chatterjee et al., 2017; Hokamp and Liu, 2017), for text simplification us-

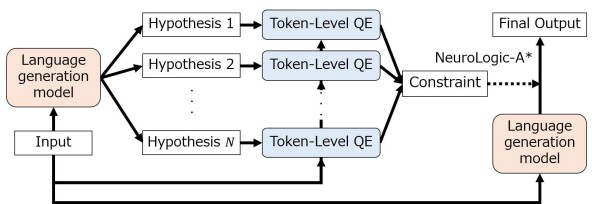

Figure 1: Overview of the proposed method

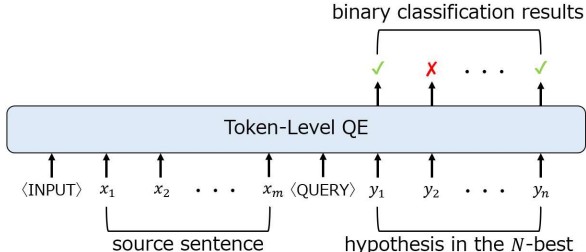

Figure 2: Token-level QE model (special input tokens are added into the vocabulary.)

| Data set | Train | Validation | Test |
|---|---|---|---|
| DIRECT | $64,126$ | – | $7,372$ |
| CNN/Daily Mail | $287,113$ | $13,368$ | $11,490$ |
| XSum | $204,045$ | $11,332$ | $11,334$ |
| COMMONGEN | $67,389$ | $4,018$ | $7,644$ |

Table 1: Number of sentences in evaluation datasets

ing difficult words as constraints (Nishihara et al., 2019; Dehghan et al., 2022; Zetsu et al., 2022), for style transfer using style-specific vocabulary as constraints (Kajiwara, 2019), and for table-to-text generation using keywords in tables as constraints (Lu et al., 2022). Different from these studies that assume the availability of task-specific knowledge, our method creates lexical constraints based purely on the $N$-best hypotheses of language generation.

We use the state-of-the-art lexically constrained decoding method, namely, NeuroLogic-A* (Lu et al., 2022). NeuroLogic-A* searches for output candidates with high generation probabilities and constraint satisfaction rates by tracking states of satisfaction by the following steps. (1) For each candidate token at a time step, NeuroLogic-A* looks ahead to future tokens to be generated. (2) Based on the look-ahead results, it computes satisfaction rates of lexical constraints and prunes the candidate tokens. (3) It groups the remaining candidates based on the states of the constraint satisfaction and selects output tokens from the best candidates in each group to preserve a broad search space.

## 3 Proposed Method

Figure 1 shows the overview of the proposed method. The main component is the token-level QE model that predicts whether each token in $N$-best hypotheses should be used or avoided in generating the final output. Tokens predicted as the former is included in *positive* constraints and those predicated as the latter are included in *negative* constraints to be considered by NeuroLogic-A*.

Specifically, we fine-tune a pretrained masked language model to conduct binary token classification as illustrated in Figure 2. For each $N$-best hypothesis of training sentences obtained by the language generation model, token-level labels are automatically assembled using their references. Hypothesis tokens appearing in the corresponding reference are labelled as positive and otherwise labelled as negative. At inference, the QE model takes the concatenation of a source and a

hypothesis in the $N$-best as input and conducts binary classification for each token. We expect that the masked language model acquires the sense of synonyms and multi-word expressions through pre-training and transfers that knowledge to our token-level QE.

Because the token-level QE is context-dependent, the same token appearing in different hypotheses may be predicted both positive and negative labels, respectively. Our model determines the final label by majority voting. If the numbers of positive and negative predictions are tie, the corresponding token is excluded from lexical constraints.

## 4 Experimental Settings

The proposed method is widely applicable to language generation tasks. We thus evaluate it on paraphrasing (§ 5.1), summarisation (§ 5.2), and constrained text generation (§ 5.3).

**Proposed Method** For each evaluation dataset (Table 1), we constructed a token-level QE model by fine-tuning a RoBERTa-base (Liu et al., 2019). Specifically, we sought the beam size of $N$ by a grid search in $[1, 5, 10, 20, 30, \cdots, 100]$ to achieve the best performance on the validation set measured by the corresponding evaluation metrics. When decoding with NeuroLogic-A* for the final output, we use the same beam size as baselines for fair comparison. For more details of the implementation, please refer to Appendix A.

|  | **Indirect-to-Direct** | | | | **Direct-to-Indirect** | | | |
| --- | --- | --- | --- | --- | --- | --- | --- | --- |
|  | w/ history | | w/o history | | w/ history | | w/o history | |
|  | BLEU | $N$ | BLEU | $N$ | BLEU | $N$ | BLEU | $N$ |
| beam-search | 35.57 | - | 34.38 | - | 26.92 | - | 26.63 | - |
| NCD (Yee et al., 2019) | $36.01^\dagger$ | 30 | $35.43^\dagger$ | 20 | $26.14^\dagger$ | 40 | $27.03^\dagger$ | 20 |
| DrNMT (Lee et al., 2021) | $35.85^\dagger$ | 30 | $34.65^\dagger$ | 60 | 27.06 | 100 | 26.50 | 20 |
| NeuroLogic-A* (P & N) | $36.43^\dagger$ | 50 | $35.42^\dagger$ | 40 | $30.21^\dagger$ | 20 | $30.57^\dagger$ | 30 |
| NeuroLogic-A* (P) | $\mathbf{36.95}^\dagger$ | 50 | $\mathbf{35.94}^\dagger$ | 40 | $\mathbf{30.89}^\dagger$ | 20 | $\mathbf{31.33}^\dagger$ | 70 |
| NeuroLogic-A* (N) | $35.84^\dagger$ | 50 | $34.82^\dagger$ | 60 | $29.97^\dagger$ | 20 | $30.12^\dagger$ | 10 |
| Reranking$_{\text{oracle}}$ | $59.71^\dagger$ | 100 | $59.16^\dagger$ | 100 | $48.95^\dagger$ | 100 | $49.25^\dagger$ | 100 |
| NeuroLogic-A* (P & N)$_{\text{oracle}}$ | $\mathbf{65.55}^\dagger$ | 100 | $\mathbf{65.31}^\dagger$ | 100 | $\mathbf{60.23}^\dagger$ | 100 | $\mathbf{60.71}^\dagger$ | 100 |
| NeuroLogic-A* (P)$_{\text{oracle}}$ | $57.85^\dagger$ | 100 | $57.38^\dagger$ | 100 | $49.42^\dagger$ | 100 | $50.15^\dagger$ | 100 |
| NeuroLogic-A* (N)$_{\text{oracle}}$ | $51.60^\dagger$ | 100 | $50.98^\dagger$ | 100 | $45.24^\dagger$ | 100 | $45.54^\dagger$ | 100 |

Table 2: Test set BLEU scores on DIRECT; $N$ determines the number of hypotheses to consider and $^\dagger$ indicates significant differences against beam-search confirmed by bootstrap resampling test (Koehn, 2004).

**Baselines** As the most basic baseline, we compared our method to beam search, of which size was borrowed from previous studies that tuned the value for underlying language generation models for each task. Wherever applicable, we also compared the strong discriminative and generative reranking methods. For the former, we used the Discriminative Reranker for Neural Machine Translation (DrNMT) (Lee et al., 2021)[1] that predicts the distribution of sentence-level evaluation metric scores given the source and $N$-best hypotheses. For the latter, we used the Noisy-Channel Decoding (NCD) (Yee et al., 2019)[2] that scores $N$-best candidates by linearly combining the probabilities of generation, target-side language model, and target-to-source generation. We trained these methods using the authors' implementations with datasets and metrics of each experiment task, where the beam sizes (the sizes of $N$) were searched in the same way as our method using the validation sets.

**Ablation** As ablation studies, we evaluated the performance of the proposed method using only positive lexical constraints (denoted as NeuroLogic-A* (P)), only negative lexical constraints (denoted as NeuroLogic-A* (N)), and both (denoted as NeuroLogic-A* (P & N)).

## 5 Experimental Results

This section discusses the experimental results. The details of implementation on each task are described in Appendix B.

### 5.1 Paraphrasing

**Setting** We used DIRECT (Direct and Indirect REsponses in Conversational Text) (Takayama et al., 2021), which provides paraphrases between indirect and direct utterances in conversation histories. We conduct both the Indirect-to-Direct and Direct-to-Indirect paraphrasing with and without the dialogue histories. Following Takayama et al. (2021), we fine-tuned BART (Lewis et al., 2020) as the underlying language generation models with the beam size of 4.

**Results** Table 2 shows the test set BLEU (Papineni et al., 2002) scores. The upper rows show the experimental results of baselines and the proposed methods using the predicted lexical constraints. In all subtasks, the proposed methods outperformed the baselines. In particular, the proposed method using only positive lexical constraints (NeuroLogic-A* (P)) achieves the best performance.

The lower rows in Table 2 show the 'oracle' performance of the proposed and reranking methods. The 'Ranking$_{\text{oracle}}$' indicates the performance when selecting the hypothesis with the highest sentence-level BLEU score against the reference. In our method, we used the 'oracle' lexical constraints obtained by accessing the references. As expected, all of the BLEU scores are much higher than the

---

[1]https://github.com/facebookresearch/fairseq/tree/main/examples/discriminative_reranking_nmt
[2]https://github.com/facebookresearch/fairseq/tree/main/examples/noisychannel

|  | CNN/Daily Mail | | XSum | |
|---|---|---|---|---|
|  | RL | $N$ | RL | $N$ |
| beam-search | 40.99 | - | 37.21 | - |
| DrNMT (Lee et al., 2021) | $40.40^{\dagger}$ | 10 | 37.18 | 10 |
| NeuroLogic-A* (P & N) | $\mathbf{41.99}^{\dagger}$ | 80 | $\mathbf{37.39}^{\dagger}$ | 90 |
| NeuroLogic-A* (P) | $41.72^{\dagger}$ | 10 | 37.16 | 10 |
| NeuroLogic-A* (N) | $41.76^{\dagger}$ | 100 | 37.24 | 90 |

Table 3: Test set ROUGE-L scores of the summarisation tasks; $N$ determines the number of hypotheses to consider and $^{\dagger}$ indicates significant differences against beam-search confirmed by approximate randomisation test (Riezler and Maxwell, 2005).

|  | CIDEr | $N$ |
|---|---|---|
| beam-search | 14.26 | - |
| NCD (Yee et al., 2019) | **16.24** | 100 |
| DrNMT (Lee et al., 2021) | 14.85 | 60 |
| SELF-CORRECT (Welleck et al., 2023) | 15.30 | - |
| +NeuroLogic (Welleck et al., 2023) | 15.28 | - |
| NeuroLogic-A* (P & N) | 15.38 | 90 |
| NeuroLogic-A* (P) | 15.62 | 10 |
| NeuroLogic-A* (N) | 14.52 | 90 |

Table 4: Test set CIDEr scores on COMMONGEN; $N$ determines the number of hypotheses to consider.

upper rows. Remarkably, NeuroLogic-A* (P & N)$_{\text{oracle}}$ largely outperforms Ranking$_{\text{oracle}}$. This result confirms that *ensembling* $N$-best hypotheses is more effective than simply selecting the best hypothesis. It supports our assumption that there exist high-quality fragments in $N$-best even though they are imperfect as the entire sentences. Moreover, the impressively higher scores under the oracle setting indicate that improving the token-level QE is a promising direction as further discussed in § 6.

## 5.2 Summarisation

**Setting** We used the CNN/Daily Mail (See et al., 2017; Hermann et al., 2015) (version 3.0.0) and XSum (The Extreme summarisation) (Narayan et al., 2018) datasets. As underlying language generation models, we used the publicly available fine-tuned BART-large models on CNN/Daily Mail and XSum released by Lewis et al. (2020) with suggested beam sizes of 4 and 6, respectively.

**Results** Table 3 shows the ROUGE-L (Lin, 2004) scores measured on the test sets. Note that NCD is not applicable to summarisation due to the unavailability of target-to-source generation model. For both CNN/Daily Mail and XSum, the proposed method using both positive and negative lexical

constraints (NeuroLogic-A* (P & N)) outperforms the baselines and achieves the highest ROUGE-L score, which has a high correlation with the human evaluation (Lin, 2004). These results confirm that the proposed method is also effective in summarisation. The full results are available in Appendix B.3.

## 5.3 Constrained Text Generation

**Setting** We used COMMONGEN (Lin et al., 2020) dataset that tasks to generate coherent sentences given a set of words. We fine-tuned GPT-2 (Radford et al., 2019) as the underlying language generation model with the beam size of 5 (Welleck et al., 2023). Different from paraphrasing and summarisation, there are a variety of possible generations as reflected in the multiple references of diverse contents. To adapt our QE model training to this task, we selected the single reference for each hypothesis that has the highest lexical overlap against the corresponding hypothesis.

**Results** Table 4 shows the CIDEr (Vedantam et al., 2015) scores measured on the test set, where SELF-CORRECT (Welleck et al., 2023) is the state-of-the-art method.[3] While our method ensembles $N$-best to improve the generation quality, SELF-CORRECT iteratively edits the initial one-best output. As the results show, our method using only positive lexical constraints (NeuroLogic-A* (P)) outperformed SELF-CORRECT, which confirms the effectiveness of ensembling high-quality fragments in the $N$-best. Nonetheless, the best method is NCD for this task. We conjecture this is because considering tokens from different hypotheses may deteriorate the generation due to the diversity in acceptable outputs. This feature is also troublesome for training discriminative reranking models as implied by the inferior performance of DrNMT. In such tasks, generative reranking models like NCD may be suitable. The full results are available in Appendix B.5.

## 6 Discussion and Future Work

As we discussed in § 5, the quality of token-level QE is critical for the performance of our method. Table 6 shows the ratio of reference tokens mistakenly included in the negative constraints ($\bar{P}_{\text{neg}}$) and the recall of positive constraints ($R_{\text{pos}}$) in the eval-

---

[3]As SELF-CORRECT is contemporaneous with our study, we borrowed these scores from the original paper. The unavailability of model outputs at the time of publication hindered further comparisons.

| Source | Well, thats like everything that is required. Thanks a lot. |
|---|---|
| Reference | Awesome. Thanks for the details. Bye. |
| Positive constraints | **Awesome**, **great**, **good**, the, details, detail, a, an, for, Have, you, day, it, . |

| Source | Chinese I think, but I need location and how to get in touch with them. |
|---|---|
| Reference | I don't mind. Maybe chinese? I need contact number and postcode. |
| Positive constraints | **phone**, **contact**, **number**, the, and, I, address, need, Chinese, Chinese, ,, . |

Table 5: Examples of constraints predicted by our method (Indirect-to-Direct transformation without history)

| Task | $\bar{P}_{\text{neg}}(\downarrow)$ | $R_{\text{pos}}(\uparrow)$ |
|---|---|---|
| Indirect-to-Direct w/ history | 0.20 | 0.41 |
| Indirect-to-Direct w/o history | 0.24 | 0.37 |
| Direct-to-Indirect w/ history | 0.23 | 0.40 |
| Direct-to-Indirect w/o history | 0.22 | 0.36 |
| CNN/Daily Mail | 0.28 | 0.31 |
| XSum | 0.24 | 0.36 |
| COMMONGEN | 0.20 | 0.36 |

Table 6: Performance of our token-level QE

uation tasks. The results indicate that 20% to 28% of reference tokens were in the negative constraints while the recalls of positive constraints were limited to 31% to 41%. Improvement of these metrics directly enhances our method, which constitutes our future work. We will explore a QE method to model interactions within and across hypotheses.

Table 5 shows examples of constraints predicted by our method on the paraphrase generation task (Indirect-to-Direct transformation without history). In the first example, synonyms of "awesome", "great", and "good" are predicted, while in the second example, multi-word expressions of "contact number" and "phone number" are predicted as positive constraints. These results indicate that our QE model preserves the ability to consider these to some extent. We should need a more sophisticated model to better handle synonyms and multi-word expressions, which constitutes our future work.

## Limitations

Our model conducts decoding twice to generate a final sentence; furthermore, the second one is lexically constrained decoding, which increases the computational cost of language generation. We measured the decoding times of the proposed and compared methods on the paraphrase generation task (Indirect-to-Direct transformation without history) under the same settings of Table 2. The programs ran on a single GPU of NVIDIA RTX A6000 with 48GB memory installed on a Linux server with 1TB memory and AMD EPYC 7552 CPU. Our naive implementation needs 1.9 sec/sent while DrNMT (Lee et al., 2021) and NCD (Yee et al., 2019) do 0.3 sec/sent on average. A straightforward remedy is to adaptively decide whether to conduct the second decoding based on the token-level QE results. For example, if there is a hypothesis of which token-level QE results imply that it satisfies a quality standard needed by a downstream task, we can directly output the hypothesis. If all the hypotheses are unsatisfactory, we can conduct the second decoding using lexical constraints.

Currently, all constraints are treated equally in lexically constrained decoding, but we assume their importance can be diverse and may change depending on the status of generation. This expansion is beyond the scope of the current paper but surely worth exploring, which constitutes our future work.

## Acknowledgements

This work was supported by JSPS KAKENHI Grant Number JP21H03564.

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

## A   Implementation Details

We implemented our token-level QE model using RoBERTa (Liu et al., 2019) with the HuggingFace Transformers (Wolf et al., 2020) library.[4] In fine-tuning RoBERTa, we calculated the F1 score on the validation set at the end of every epoch and stopped tuning when there was no improvement for 3 epochs.

We used the implementation of Lu et al. (2022) to replicate NeuroLogic-A*.[5] However, due to the lack of negative lexical constraints in the original implementation, we modified the codes to allow negative lexical constraints.

## B   Experiment Details

### B.1   Paraphrasing Experiment

The **DIRECT** corpus is an extension of the multi-domain, multi-turn, task-oriented dialogue corpus of MultiWOZ 2.1 (Multi-Domain Wizard-of-Oz 2.1) (Budzianowski et al., 2018; Eric et al., 2020). DIRECT provides the dialogue histories in Multi-WOZ, the original responses, indirect paraphrases of the original responses, and direct paraphrases of the original responses.

We fine-tuned a 'facebook/bart-base'[6] model using the HuggingFace Transformers library with the same setting as Takayama et al. (2021). The beam size was set to 4 following the experiments in the original paper.

### B.2   Summarisation Experiment

The **CNN/Daily Mail** dataset is a collection of CNN and Daily Mail articles and highlights (summaries), and consists of about 310k news articles and highlight pairs. The average number of sentences in the CNN/Daily Mail dataset is 30.7 for articles and 3.8 for highlights. The **XSum** dataset is a collection of BBC articles and their summaries and consists of about 230k article-summary pairs. The average number of sentences in XSum is 19.8 for

articles and 1.0 for summaries. The XSum dataset requires less number of summary sentences than the CNN/Daily Mail dataset; therefore, it requires more abstract summarisation. The maximum input length of our token-level QE model is 512. If an input length exceeds that limit, we split the article into two and input to the model, and then merge the prediction results.

As underlying language generation models for summarisation, we used 'facebook/bart-large-cnn'[7] and 'facebook/bart-large-xsum'[8]. These models have been fine-tuned on CNN/Daily Mail and XSum datasets, respectively. Their beam sizes are suggested as 4 and 6, respectively.

### B.3   Summarisation Results

Table 7 shows test set results of all evaluation metrics. The bottom three rows indicate the performance when using the oracle lexical constraints created by accessing references.

### B.4   Constrained Text Generation Experiment

The COMMONGEN dataset consists of $35,141$ concept sets associated with $77,449$ sentences. The average length of reference sentences in the COMMONGEN dataset is $10.86$.

We fine-tuned a 'gpt2-large'[9] model with the same setting as Lin et al. (2020). The evaluation metrics were computed using the official script[10].

### B.5   Constrained Text Generation Results

Table 8 shows test set results of all evaluation metrics. The bottom rows present the results under the oracle setting. Different from paraphrasing and summarisation, the oracle reranking, which chooses a hypothesis with the highest evaluation score, outperformed our methods with oracle lexical constraints. Our manual investigation confirmed that the references of a source sentence are diverse in COMMONGEN, and thus considering tokens from different references can be harmful. This result implies that the diversity in possible generations affects the performance of the proposed method.

---

[4]https://huggingface.co/roberta-base
[5]https://github.com/GXimingLu/a_star_neurologic
[6]https://huggingface.co/facebook/bart-base

[7]https://huggingface.co/facebook/bart-large-cnn
[8]https://huggingface.co/facebook/bart-large-xsum
[9]https://huggingface.co/gpt2-large
[10]https://github.com/INK-USC/CommonGen

| | CNN/Daily Mail | | | | XSum | | | |
|---|---|---|---|---|---|---|---|---|
| | R1 | R2 | RL | $N$ | R1 | R2 | RL | $N$ |
| beam-search | 44.04 | 21.08 | 40.99 | - | 45.46 | 22.35 | 37.21 | - |
| DrNMT (Lee et al., 2021) | 43.53[†] | 20.62[†] | 40.40[†] | 10 | 45.37 | 22.33 | 37.18 | 10 |
| NeuroLogic-A* (P & N) | **44.99**[†] | 21.64[†] | **41.99**[†] | 80 | 45.69[†] | 22.37 | **37.39**[†] | 90 |
| NeuroLogic-A* (P) | 44.76[†] | 21.33[†] | 41.72[†] | 10 | **45.87**[†] | 22.19[†] | 37.16 | 10 |
| NeuroLogic-A* (N) | 44.72[†] | **21.72**[†] | 41.76[†] | 100 | 45.18[†] | 22.25 | 37.24 | 90 |
| Reranking$_{oracle}$ | 53.78[†] | 21.64[†] | 41.99[†] | 100 | 56.74[†] | 35.19[†] | 51.96[†] | 100 |
| NeuroLogic-A* (P & N)$_{oracle}$ | **61.74**[†] | **33.84**[†] | **54.75**[†] | 100 | **67.23**[†] | **42.72**[†] | **54.69**[†] | 100 |
| NeuroLogic-A* (P)$_{oracle}$ | 56.05[†] | 30.58[†] | 52.05[†] | 100 | 61.22[†] | 35.68[†] | 47.52[†] | 100 |
| NeuroLogic-A* (N)$_{oracle}$ | 53.30[†] | 29.89[†] | 50.09[†] | 100 | 55.33[†] | 32.88[†] | 47.14[†] | 100 |

Table 7: Test set ROUGE scores of the summarisation tasks; $N$ determines the number of hypotheses to consider and [†] indicates significant differences against beam-search confirmed by approximate randomisation test (Riezler and Maxwell, 2005).

| | BLEU-4 | CIDEr | Coverage | $N$ |
|---|---|---|---|---|
| beam-search | 27.08 | 14.26 | 84.48 | - |
| NCD (Yee et al., 2019) | **31.52** | **16.24** | 91.73 | 100 |
| DrNMT (Lee et al., 2021) | 27.55 | 14.85 | 91.73 | 60 |
| SELF-CORRECT (Welleck et al., 2023) | 27.98 | 15.30 | 94.58 | - |
| SELF-CORRECT+NeuroLogic (Welleck et al., 2023) | 28.17 | 15.28 | **97.80** | - |
| NeuroLogic-A* (P & N) | 28.85 | 15.38 | 91.39 | 90 |
| NeuroLogic-A* (P) | 28.04 | 15.62 | 94.06 | 10 |
| NeuroLogic-A* (N) | 27.41 | 14.52 | 86.71 | 90 |
| Reranking$_{oracle}$ | **52.70** | **21.62** | 90.15 | 100 |
| NeuroLogic-A* (P & N)$_{oracle}$ | 42.51 | 19.20 | 96.71 | 100 |
| NeuroLogic-A* (P)$_{oracle}$ | 38.52 | 18.98 | **97.81** | 100 |
| NeuroLogic-A* (N)$_{oracle}$ | 30.66 | 15.29 | 86.99 | 100 |

Table 8: Test set scores on COMMONGEN; $N$ determines the number of hypotheses to consider.