# OpenReview forum: "Self-Ensemble of $N$-best Generation Hypotheses by Lexically Constrained Decoding"
_EMNLP/2023/Conference — EMNLP 2023 Main_

### Official Review · Reviewer_26qx · 2023-08-02

**Soundness:** 3
**Typos Grammar Style And Presentation Improvements:** 1. in line 030, raking -> ranking

**Excitement:**

3: Ambivalent: It has merits (e.g., it reports state-of-the-art results, the idea is nice), but there are key weaknesses (e.g., it describes incremental work), and it can significantly benefit from another round of revision. However, I won't object to accepting it if my co-reviewers champion it.

**Paper Topic And Main Contributions:**

This paper is about how to combine N-best generations from beam search to construct the final generation.

To this end, they first train a token-level quality estimation (QE) module to predict whether each token in the N-best hypotheses should appear in the final generation. Then based on the outputs of this QE component, they use lexically constrained decoding to produce the final generation.

Experiments on paraphrasing generation, summarization, and CommonGen demonstrate that their method improves the generation performance compared to beam search and a few other reranking baselines.

The contribution of this work is made toward the NLP engineering experiment direction.

**Questions For The Authors:**

Question A: I don't understand lines 126 to 128. If labels are assigned based on their appearance in references, why the same token in different hypotheses can have different labels?

Question B: For DrNMT and NCD, did you search the beam size as you did for your method?

**Reasons To Accept:**

1. The idea of combining N-best hypotheses is reasonable, and the proposed method is straightforward.

2. Experiments are done on multiple generation tasks which demonstrate the generalization ability of the proposed method.

**Reasons To Reject:**

1. The proposed QE component is token-level, which can not take compositional semantics into consideration. And the label of each token is determined by whether it appears in the reference, which is quite rigid and can not take synonyms into consideration.

2. The beam size is searched for the proposed method (line 142), while the default setting is used for standard beam search (line 150).

3. Significance test is missing when "our methods outperformed SELF-CORRECT" (line 251) is claimed while only 0.1 point improvement is observed in table 4.

**Reproducibility:**

4: Could mostly reproduce the results, but there may be some variation because of sample variance or minor variations in their interpretation of the protocol or method.

**Reviewer Confidence:**

4: Quite sure. I tried to check the important points carefully. It's unlikely, though conceivable, that I missed something that should affect my ratings.

---

> ### Author Rebuttal · Authors · 2023-08-29
>
> We appreciate your detailed comments on our paper. First of all, we would like to clarify the process for constructing our QE model, which may be useful for addressing some of the misunderstandings. Our method *trains* token-level quality estimation model by fine-tuning RoBERTa, where the references of a training corpus are used to generate ground-truth labels. After training, the QE model *predicts* which tokens in an output candidate (of N-best) should be or should not be used in the final output referring to the context within. Therefore, our QE model is NOT dependent on nor constrained by references at inference time.
>
>
> **Reasons To Reject 1-1:** The proposed QE component is token-level, which can not take compositional semantics into consideration.
>
> **Response:** As our QE model is based on RoBERTa (see lines 138-140), it can take the entire sentence as context for prediction, which would allow considerations of compositional semantics to some extent, even if being imperfect. We totally agree that the consideration of compositional semantics is crucial direction and worth exploring, which constitutes our future work. We would like to note that NeuroLogic-A* allows inclusion of set constraints, and thus we can easily apply compositional constraints when they are ready.
>
>
> **Reasons To Reject 1-2:** And the label of each token is determined by whether it appears in the reference, which is quite rigid and can not take synonyms into consideration.
>
> **Response:** We respectfully point out that our QE model should acquire the ability to consider synonyms through both pre-training of RoBERTa and fine-tuning thereafter of which corpus is created from a large training corpus of the corresponding text generation task.
>
>
> **Reasons To Reject 2:** The beam size is searched for the proposed method (line 142), while the default setting is used for standard beam search (line 150).
>
> **Response:** We would clarify that the “default setting” was used for *both* of our method and the standard beam search for generating final outputs. What we tuned for our method was the size of N, which affects only the creation of lexical constraints by determining how many candidates to input to the QE model to generate lexical constraints (see lines 141-146).
>
>
> **Reasons To Reject 3:** Significance test is missing when "our methods outperformed SELF-CORRECT" (line 251) is claimed while only 0.1 point improvement is observed in table 4.
>
> **Response:** SELF-CORRECT (Welleck et al., 2023) was published on May 2023, which should be considered as contemporaneous to our submission (EMNLP2023 submission guideline regulates that such contemporaneous works should not be considered as targets for detailed comparisons that require additional experimentation and/or in-depth analysis). Furthermore, the unavailability of outputs released by the authors hindered us from conducting significance tests. While our NeuroLogic-A* (P) has 0.32 higher point to SELF-CORRECT, we will revise the corresponding description in the final version if the paper is accepted.
>
>
> **Questions For The Authors (Question A):** I don't understand lines 126 to 128. If labels are assigned based on their appearance in references, why the same token in different hypotheses can have different labels?
>
> **Response:** Lines 126-128 describe the process in inference time, where the labels are *predicted* by our QE model (remind that references are only used for training the QE model). Because our QE model is context-dependent, the same token appearing in different sentences of N-best may be predicted different labels.
>
>
> **Questions For The Authors (Question B):** For DrNMT and NCD, did you search the beam size as you did for your method?
>
> **Response:** Yes, we searched the beam size of these methods (the sizes of N)  in the same way as our method using validation sets. We will clarify this in the final version if the paper is accepted.
>
>
> **Typos Grammar Style And Presentation Improvements:** in line 030, raking -> ranking
>
> **Response:** Thank you for pointing this out. We will correct it in the camera-ready version.

---

### Official Review · Reviewer_8gyk · 2023-08-04

**Soundness:** 4

**Excitement:**

4: Strong: This paper deepens the understanding of some phenomenon or lowers the barriers to an existing research direction.

**Paper Topic And Main Contributions:**

This paper proposes a novel post-processing method for text generation tasks.
After beam search generation, this method is able to combine the high-quality parts in the N-best hypothesis and re-generate a better answer.
Experiments on several text generation tasks show the effectiveness of this method.

**Reasons To Accept:**

1. It is an interesting idea to combine the token level quality estimation and lexically constrained decoding to improve the generation quality.

2. Experiments on several tasks demonstrate the effectiveness of the proposed method.

3. The paper is well organized and easy to follow, although more proofreading might be required.

**Reasons To Reject:**

Although the author mentioned in the Limitations section that this method is more time-consuming compared to reranking, it is better to provide a time comparison in the paper for reference in subsequent studies.

**Reproducibility:**

4: Could mostly reproduce the results, but there may be some variation because of sample variance or minor variations in their interpretation of the protocol or method.

**Reviewer Confidence:**

4: Quite sure. I tried to check the important points carefully. It's unlikely, though conceivable, that I missed something that should affect my ratings.

**Typos Grammar Style And Presentation Improvements:**

This paper might require more proofreading:

1. summarisation --> summarization
2. L030 raking --> ranking

---

> ### Author Rebuttal · Authors · 2023-08-29
>
> Thank you for your great efforts in reviewing. We appreciate your acknowledging the contributions of our study.
>
> **Reasons To Reject:** Although the author mentioned in the Limitations section that this method is more time-consuming compared to reranking, it is better to provide a time comparison in the paper for reference in subsequent studies.
>
> **Response:** Thank you for the valuable suggestion. We measured the decoding time of our and compared methods on paraphrase generation (Indirect-to-Direct transformation without history) under the same settings of Table 2. Our naive implementation needs 1.9 sec / sent while DrNMT (Lee et al., 2021) and NCD (Yee et al., 2019) do 0.3 sec / sent on average. We believe the computational efficiency will be improved by more professional engineering for sure. We will expand the Limitation section to include these results in the final version if the paper is accepted.
>
>
> **Typos Grammar Style And Presentation Improvements:** This paper might require more proofreading:
> summarisation --> summarization
> L030 raking --> ranking
>
> **Response:** Thank you for pointing these out. We will revise the paper with careful proofreading in the final version (for “summarisation,” we used UK spelling).

---

### Official Review · Reviewer_5Mk4 · 2023-08-12

**Soundness:** 3

**Excitement:**

3: Ambivalent: It has merits (e.g., it reports state-of-the-art results, the idea is nice), but there are key weaknesses (e.g., it describes incremental work), and it can significantly benefit from another round of revision. However, I won't object to accepting it if my co-reviewers champion it.

**Paper Topic And Main Contributions:**

This paper introduces a natural language generation method, which merges the high-quality fragments in N-best hypotheses and applies tokens that should or should not be present in the final output as lexical constraints in decoding. The proposed method is actually a N-best ensemble method for improving the quality of language generation. And it can be easily to apply to a variety of language generation tasks such as paraphrasing, summarization, and constrained text generation.

**Reasons To Accept:**

The proposed method in this paper is intuitive and effective through some experiments on paraphrasing, summarization, and constrained text generation tasks.

**Reasons To Reject:**

The proposed method needs to conduct decoding twice to generate a final sentence, which is inefficient and is not suitable for practical application.

**Reproducibility:**

4: Could mostly reproduce the results, but there may be some variation because of sample variance or minor variations in their interpretation of the protocol or method.

**Reviewer Confidence:**

4: Quite sure. I tried to check the important points carefully. It's unlikely, though conceivable, that I missed something that should affect my ratings.

---

> ### Author Rebuttal · Authors · 2023-08-29
>
> Thank you for your time and efforts in reviewing our paper.
>
> **Reasons To Reject:** The proposed method needs to conduct decoding twice to generate a final sentence, which is inefficient and is not suitable for practical application.
>
> **Response:** We agree that our implementation may not be immediately applicable to “online” systems. As we experimented, current naive implementation needs about 1.9 sec / sent for decoding (for details, see our response to Reviewer 8gyk). The performance should be improved by more professional engineering for sure. In addition, our method has a wide applicability to various language generation tasks (and indeed achieved remarkable improvements on three of them), which can be used in “offline” scenarios where “online” responses are not needed.  More importantly, computational efficiency should not be the only reason for avoiding exploring promising technologies, which would hinder technical breakthroughs in the future.

---

### Meta-Review · Area_Chair_s9Tk · 2023-09-17

**Recommendation:** 4

**Metareview:**

This paper presents an interesting approach that recycles parts of hypotheses in N-best lists produced via beam search to ultimately generate a higher-quality response. The method is evaluated on paraphrase generation, summarization, and commonsense generation and yields improved performance compared to beam search and other baselines. All reviewers praise the idea behind the method, as it is intuitive and clever to combine token-level quality estimation with constrained decoding. Criticisms of the approach include the computational expense (the authors provide some timing comparisons in their response to 8gyk, but they include no details on experimental/hardware settings so it is hard to draw meaningful conclusions from that). There were other concerns about experimetnal conditions (e.g. beam size) that were mostly resolved through discussion. Overall, it seems like a solid paper, but I would hope the authors provide a thorough analysis in the next version of how much more expensive this approach is in terms of time compared to normal beam search with a variety of batch sizes.

---

### Decision · Program_Chairs · 2023-10-07

**Decision:**

Accept-Main

**Comment:**

This paper presents an interesting approach that recycles parts of hypotheses in N-best lists produced via beam search to ultimately generate a higher-quality response. The method is evaluated on paraphrase generation, summarization, and commonsense generation and yields improved performance compared to beam search and other baselines. All reviewers praise the idea behind the method, as it is intuitive and clever to combine token-level quality estimation with constrained decoding. Criticisms of the approach include the computational expense (the authors provide some timing comparisons in their response to 8gyk, but they include no details on experimental/hardware settings so it is hard to draw meaningful conclusions from that). There were other concerns about experimetnal conditions (e.g. beam size) that were mostly resolved through discussion. Overall, it seems like a solid paper, but I would hope the authors provide a thorough analysis in the next version of how much more expensive this approach is in terms of time compared to normal beam search with a variety of batch sizes.